# Evolution of MicroRNA Biogenesis Genes in the Sterlet (*Acipenser ruthenus*) and Other Polyploid Vertebrates

**DOI:** 10.3390/ijms21249562

**Published:** 2020-12-15

**Authors:** Mikhail V. Fofanov, Dmitry Yu. Prokopov, Heiner Kuhl, Manfred Schartl, Vladimir A. Trifonov

**Affiliations:** 1Institute of Molecular and Cellular Biology SB RAS, Lavrentiev Ave. 8/2, 630090 Novosibirsk, Russia; dprokopov@mcb.nsc.ru; 2Department of Natural Sciences, Novosibirsk State University, Pirogova 2, 630090 Novosibirsk, Russia; 3Leibniz-Institute of Freshwater Ecology and Inland Fisheries, Müggelseedamm 301 and 310, 12587 Berlin, Germany; kuhl@igb-berlin.de; 4Developmental Biochemistry, Biocenter, University of Wuerzburg, Am Hubland, 97074 Wuerzburg, Germany; phch1@biozentrum.uni-wuerzburg.de; 5Xiphophorus Genetic Stock Center, Texas State University, 601 University Drive, 419 Centennial Hall, San Marcos, TX 78666-4616, USA

**Keywords:** sturgeon, whole genome duplication, microRNA, gene duplications

## Abstract

MicroRNAs play a crucial role in eukaryotic gene regulation. For a long time, only little was known about microRNA-based gene regulatory mechanisms in polyploid animal genomes due to difficulties of polyploid genome assembly. However, in recent years, several polyploid genomes of fish, amphibian, and even invertebrate species have been sequenced and assembled. Here we investigated several key microRNA-associated genes in the recently sequenced sterlet (*Acipenser ruthenus*) genome, whose lineage has undergone a whole genome duplication around 180 MYA. We show that two paralogs of *drosha*, *dgcr8*, *xpo1*, and *xpo5* as well as most *ago* genes have been retained after the acipenserid-specific whole genome duplication, while *ago1* and *ago3* genes have lost one paralog. While most diploid vertebrates possess only a single copy of *dicer1*, we strikingly found four paralogs of this gene in the sterlet genome, derived from a tandem segmental duplication that occurred prior to the last whole genome duplication. *ago1,3,4* and *exportins1,5* look to be prone to additional segment duplications producing up to four-five paralog copies in ray-finned fishes. We demonstrate for the first time exon microsatellite amplification in the acipenserid *drosha2* gene, resulting in a highly variable protein product, which may indicate sub- or neofunctionalization. Paralogous copies of most microRNA metabolism genes exhibit different expression profiles in various tissues and remain functional despite the rediploidization process. Subfunctionalization of microRNA processing gene paralogs may be beneficial for different pathways of microRNA metabolism. Genetic variability of microRNA processing genes may represent a substrate for natural selection, and, by increasing genetic plasticity, could facilitate adaptations to changing environments.

## 1. Introduction

The hypothesis of animal genome evolution by whole genome duplications (later known as “the 2R hypothesis”) was first suggested by Susumu Ohno 50 years ago [1]. According to this theory, all extant vertebrates are derived from a common ancestor, which experienced two rounds of whole-genome duplication (1R and 2R of WGD) over 500 million years ago (MYA) [2,3]. Some taxonomic groups of animals went through one or more additional rounds of WGD. Ray-finned fishes (Actinopterygii) are the most speciose taxonomic group of vertebrates and several additional WGDs have been revealed in different fish lineages. Thus, the common ancestor of teleosts (the largest infraclass of ray-finned fishes) experienced a WGD (teleost-specific 3R) around 320 MYA [4,5,6]. In addition, some teleost lineages such as salmonids and carps have experienced an additional WGD (4R) [7,8,9]. Besides ray-finned fishes, polyploid species occur in other vertebrate groups such as amphibians, e.g., the African clawed frog went through a 3R WGD event about 48 MYA [10].

Studies of polyploid genomes have long been complicated by the difficulty of polyploid genome assembly. Recently, several polyploid animal genome assemblies have been published, including sterlet (*Acipenser ruthenus*), goldfish (*Carassius auratus*), common carp (*Cyprinus carpio*), Atlantic salmon (*Salmo salar*), rainbow trout (*Oncorhynchus mykiss*), African clawed frog (*Xenopus laevis*), bdelloid rotifer (*Adineta vaga*), common house spider (*Parasteatoda tepidariorum*), Arizona bark scorpion (*Centruroides sculpturatus*), and mangrove horseshoe crab (*Carcinoscorpius rotundicauda*) [7,8,9,10,11,12,13,14,15].

Besides whole genome duplications, segmental duplications (predominantly tandemly arranged) also can take place and increase gene copy numbers [16]. Gene duplications represent the major driving force in the evolution of vertebrates [17]. As both WGD and segmental duplications generate functional redundancy, the duplicated copies can follow different evolutionary ways, which are generally restricted to only four available routes: coexpression (both copies retain their function), nonfunctionalization (function loss or complete deletion of one copy), subfunctionalization (specialization of each copy, subfunction partition), and neofunctionalization (acquisition of a novel function).

Although the consequences of gene and genome duplications for different protein coding genes have been studied widely, the evolution of genes involved in the biogenesis of small RNA is still unexplored. Small RNAs are short (≈18–30 nucleotides) noncoding RNA molecules that play a key role in post-transcriptional silencing of target RNAs in different eukaryotic lineages [18]. Three classes of small RNAs that are involved in RNA silencing or post-transcriptional regulation of gene expression have been described: microRNAs (miRNAs), short interfering RNAs (siRNAs), and PIWI-associated RNAs (piRNAs). In addition to structural and expression features, various small RNAs differ in their functions: miRNAs are involved in the regulation of gene expression, siRNAs protect the host organism from the invasion of viruses, and piRNAs protect the germline cells from excessive reproduction of transposons [19].

Transcription of miRNA genes results in long primary miRNAs (pri-miRNAs) that are first processed into ≈70 bp long precursor miRNAs (pre-miRNAs), and then to mature miRNAs. Originally described in *Drosophila*, Drosha and Pasha (the vertebrate ortholog of Pasha is called DGCR8—DiGeorge syndrome chromosomal region) are the key proteins of the microprocessor complex, which is involved in the initiation of miRNAs processing in the nucleus. This complex catalyzes double-stranded RNA (dsRNA) hairpin cleavage during the first processing step of miRNAs. The *Drosha* gene encodes the Drosha enzyme belonging to the class 2 of RNase III family. The *Pasha* gene encodes the double-stranded RNA binding protein. Pasha/DGCR8 acts together with Drosha within the microprocessor nuclear complex, which is required to convert pri-miRNA to pre-miRNA. Previously, it was widely believed that the presence of the microprocessor protein machinery and the miRNA pathway in total are distinctive features of the animal kingdom [20,21]. However, Drosha and Pasha homologs were found recently in several early-branching lineages such as *Ichthyosporea*, supporting the hypothesis of the unicellular pre-metazoan origin of the microRNA machinery [22]. Generally, both *Drosha* and *Pasha* are single-copy genes in vertebrates despite two WGD events in the common vertebrate ancestor [23].

Exportin 5 (Xpo5) is one of the key nuclear transporters of pre-miRNA from the nucleus to the cytoplasm, which performs its function predominantly in a Ran-GTF-dependent manner [24,25,26]. It was previously shown that besides Xpo5, Exportin 1 (Xpo1) (which is usually required to transport snRNAs) can also be involved in pre-miRNA transport, especially in a quiescent or growth-arrested cellular state, where *Xpo5* expression is repressed [27]. Experiments with *Xpo5* knockouts confirmed the existence of some alternative mechanisms of pre-miRNA export to the cytoplasm because of its lower impact on miRNA expression suppression than the more effective *Dicer* or *Drosha* knockouts [28].

The *Dicer1* gene encodes a helicase with RNase motif. Dicer1 together with Drosha belongs to the RNase III family [29]. Dicer cleaves long double-stranded RNA (dsRNA) and pre-miRNA in the cytoplasm producing short single-stranded RNA fragments—small interfering RNAs and microRNAs. Dicer1 activates the RNA-induced silencing complex (RISC, with a catalytic component—Argonaute), and thus it is an essential component of the RNA interference machinery [30]. Dicer proteins were found in different taxonomic groups including animals, fungi, and plants. Previously, it was shown that representatives of these groups may contain not only a single copy of the *Dicer* gene but encode several copies retained after *Dicer* duplication events [31]. It was hypothesized that a «proto-*Dicer*» was present in early multicellular organisms and subsequent *Dicer* duplications resulted in the emergence of the *Dicer* family. Thus, fungal genomes generally encode two Dicer proteins (Dicers alpha and beta) and plant genomes encode four copies (Dicer-like 1–4 or DCL1–4). The number of Dicer-encoding genes in metazoans varies from a single copy in nematodes and most vertebrates (*Dicer1*) and two in invertebrates such as insects and flatworms (*Dicer1* and *Dicer2*) to up to five copies in the primitive placozoan *Trichoplax adhaerans* (all from *Dicer2* subfamily) [32]. It was also shown that the *Dicer* family was lost in many parasitic protozoa and RNAi-lacking fungi such as *Saccharomyces cerevisiae* [33]. Earlier, it was widely believed that *Dicer2* in insects has resulted from an insect-specific *Dicer* duplication [31], but later research revealed that a *Dicer1/2* duplication event more likely occurred very early in the metazoan evolution as an ancient duplication [29]. Thus, two Dicer proteins have been found in non-insect invertebrates: in the crustacean *Litopenaeus vannamei* and in two planarians (*Clonorchis sinensis* and *Schmidtea mediterranea)*, where one of these Dicers grouped with insect Dicer1 and another with Dicer2. *Dicer2* orthologs were also identified in the basal metazoa, *Trichoplax* and *Nematostella*, supporting this hypothesis [29]. *Dicer* family members differ not only in copy number but also functionally and structurally. It is well known that Dicer1 in insects is involved in miRNA-mediated gene regulation and Dicer2 in antiviral immunity [34,35,36]. Plant Dicer-like protein DCL4 was also found to be associated with antiviral immunity [37,38,39]. DCL1 is the only plant Dicer protein that produces miRNAs, while DCL2–4 are involved in siRNA-mediated silencing [40]. Monocots have additional DCL5, which is specifically expressed in reproductive organs, thus it might perform a similar role as a piRNA pathway in vertebrates to suppress transposons in the germline [40].

The metazoan Argonaute protein superfamily was traditionally grouped into AGO and PIWI Argonautes. Recent phylogenetic analyses showed the existence of three conserved Argonaute classes, namely siRNA-class AGO, miRNA-class AGO, and PIWI Argonautes. It was also shown that vertebrates lack siRNA-class AGO proteins and the retained vertebrate AGOs have low rates of molecular evolution [41]. PIWI Argonautes are involved in silencing of transposable elements and interact specifically with piRNAs. The diversity of Argonaute proteins results from both whole genome and gene duplication events producing different RNAi-like mechanisms in metazoans and plants [42,43].

*Argonaute* and *Dicer* genes are closely associated in evolution. This fact is supported by research demonstrating that these genes were often acquired and lost together in several lineages and their functions are tightly linked [44]. It was shown that a *Dicer1/2* duplication occurred at the same time as the *Ago1/2* duplication event after sponges diverged from the main metazoan tree but before the divergence of cnidaria [29].

In general, vertebrates contain four types of Argonaute proteins of the miRNA-class: Ago1, Ago2, Ago3, and Ago4, which are usually encoded by single-copy genes. However, there are some exceptions such as some groups which have undergone WGD events and retained their polyploidy status. In the latter case, every gene belonging to the metazoan Argonaute superfamily can be duplicated [45].

The Acipenseriformes represent one of the basal orders of ray-finned fishes and are regarded as “living fossils” with a history of more than 200 million years [46]. Previously, it was shown that all representatives of the order are polyploids, and in some lineages the process is still ongoing and groups with higher ploidy levels are common [47]. Recently, the genome of an acipenseriform species, the sterlet (*Acipenser ruthenus*), was sequenced and assembled [11], which makes it possible to analyze its gene content and compare it to other vertebrate species. As a polyploid non-teleost species of ray-finned fishes (the last WGD event took place 180 My ago), sterlet is very interesting for comparative genomic studies on the evolution of microRNA processing machinery. Here we investigated the copy number, gene structure, and transcription of *dicer1*, *drosha*, *dgcr8*, *xpo5*, *xpo1*, *ago1*, *ago2*, *ago3*, *ago4*, *piwi-like1,* and *piwi-like2* in the sterlet genome. We discovered that most gene duplicates have been retained after the Ac3R WGD and transcription analysis indicated active transcription of both paralogs, although some expression divergence between some paralogs was also observed.

## 2. Results

The results of the sterlet RNA-processing gene paralog identification and comparison are briefly summarized in Table 1. Noteworthy, most paralogous genes are localized on paralogous chromosomes, further confirming their origin from the last genome duplication event (Ac3R WGD). The intron-exon structure and protein composition are highly conserved.

### 2.1. Paralogs of Drosha

We found two different paralogous *drosha* genes in the sterlet genome. These genes, *drosha1* and *drosha2*, encode proteins of similar size and composition and are located on the paralogous sterlet chromosomes ARUT3 and ARUT4, respectively (Table 1). An analysis of the *drosha2* gene and its protein product revealed that the CDS contains a 54 bp-long insertion in the 4th exon, which was derived from an expansion of the GAGAGG hexanucleotide (11× amplification) and can be translated into (ER)_11_ amino acid sequence (Figure 1).

Phylogenetic analysis showed that *drosha* has been retained in two copies in some other recent polyploid vertebrates (*Carassius auratus* and *Xenopus laevis*) too, while in *Cyprinus carpio*, *Salmo salar*, and *Oncorhynchus mykiss* we found only a single copy. Noteworthy, only a single copy of the gene was found in most teleosts despite the teleost-specific genome duplication (Figure 2).

Transcriptome analysis revealed that both *drosha* copies are transcriptionally active, with *drosha2* showing higher expression in ovary, spleen, and undifferentiated gonads (Figure 3).

### 2.2. Paralogs of dgcr8

Our analysis revealed two paralogous genes encoding dgcr8 in the sterlet genome. Both genes, *dgcr8_1* and *dgcr8_2*, have almost identical amino acid sequences and are located on paralogous sterlet chromosomes ARUT12 and ARUT19. Both *dgcr8* copies are transcriptionally active with a similar expression level across different tissues (Figure 3).

Phylogenetic analysis showed that both copies of *dgcr8* have been retained in relatively recent polyploids (carps, salmonids, and African clawed frog) (Figure 4). However, genomic analysis of paralog localization in *Xenopus laevis* reveals that these copies are tandemly arranged on the same chromosome, assuming additional segmental duplication on one paralogous chromosome (Xla1L) and loss of the gene on the other paralog (Xla1S).

Both copies of *dgcr8* are actively transcribed in different sterlet organs with higher expression in ovary, spleen, and testis (Figure 3).

### 2.3. Paralogs of dicer1

Our analysis of the sterlet genome revealed the presence of four different paralogs of *dicer1—dicerA_1*, *dicerA_2*, *dicerB_1*, and *dicerB_2*.

The genes *dicerA_1* and *dicerB_1* are tandemly arranged and reside on sterlet chromosome ARUT24. Despite early divergence (preceding the Ac3R WGD), dicerA_1 and dicerB_1 protein products show high (99.37%) similarity.

Sterlet chromosome ARUT16 (paralogous to ARUT 24) contains a paralog of *dicerB_1—dicerB_2*. The current sterlet assembly does not show the presence of the *dicerA_2* paralog on ARUT 16, although the duplicated *clmnb2* is located on ARUT16, but *dicerA_2* can be found on an unplaced short scaffold (NW_023260698.1, 157,476 bp). It seems plausible that this scaffold will be assigned to ARUT16 as well (Figure 5).

*dicerB_1* and *dicerB_2* are much more similar than *dicerA_1* and *dicerB_1*, have identical exon-intron structure (28 exons) and encode only one transcript variant per gene, XM_034927235.1 (6125 bp) and XM_034035508.2 (10302 bp), which can be translated into 1896 aa proteins.

*dicerA_1* and *dicerA_2* contain 29 exons and encode two transcript variants per gene: XM_034037909.2 (6738 bp) and XM_034037910.2 (6870 bp) from *dicerA_1*, XM_034915798.1 (6652 bp) and XM_034915799.1 (6817 bp) from *dicer A_2*. Despite the difference in mRNA length, these transcript variants can be translated into 100% identical 1894 aa-long proteins.

The phylogenetic analysis demonstrates that the pairs from different paralogous chromosomes (*dicerA_1/dicerA_2* and *dicerB_1/dicerB_2*) are more similar than the pairs from the same synteny group (*dicerA_1/dicerB_1* and *dicerA_2/dicerB_2*) (Figure 6), which confirms their origin from an ancient segmental duplication. Recent polyploids (carps, salmonids, and *Xenopus laevis*) have retained only two paralogous copies.

Transcription analysis revealed that all four *dicers* are transcribed at similar levels in different tissues.

### 2.4. Paralogs of Exportin 5

Our analysis revealed two paralogous genes encoding xpo5 in the sterlet genome. These genes, *xpo5_1* and *xpo5_2*, containing 32 and 35 exons and encoding 1209 aa long proteins, respectively, are located on the paralogous sterlet chromosomes ARUT5 and ARUT6. Both *xpo5* copies are transcriptionally active, but *xpo5_1* is expressed higher in almost all studied organs, while *xpo5_2* is less active (Figure 3).

Phylogenetic analysis showed the presence of two *xpo5* paralogs in salmonids, but only a single copy was found in carps. Noteworthy is the presence of three *xpo5* paralogs in both diploid *Xenopus tropicalis* and tetraploid *Xenopus laevis* (Figure 7).

### 2.5. Paralogs of Exportin 1

Our analysis revealed two paralogous *xpo1* genes in the sterlet genome. These genes, *xpo1_1* and *xpo1_2*, containing 25 and 26 exons and encoding 1071 aa-long proteins, are located on the paralogous sterlet chromosomes ARUT5 and ARUT6. *xpo1_2* encodes another N-terminal truncated 940 aa long protein variant (XP_033873784.1) due to the presence of an alternative start codon.

Expression analysis showed that both *xpo1* paralogs are transcribed, but *xpo1_2* is significantly more active in all studied tissues, while *xpo1_1* is coexpressed at the same level in ovary (Figure 3).

Phylogenetic reconstruction demonstrated the presence of two paralogs in both diploid (*Danio rerio*, *Oryzias latipes*, *Takifugu rubripes*) and tetraploid (*Salmo salar* and *Xenopus laevis*) species, but 3–5 copies in carps and, surprisingly, four *xpo1* genes in *Oncorhynchus mykiss* (Figure 8). Incongruence in phylogenetic tree results from the presence of multiple paralogs in teleosts.

### 2.6. Paralogs of Argonaute Genes

Our analysis of sterlet genome revealed the presence of two different paralogs of the gene encoding argonaute-2 protein: *ago2_1* and *ago2_2*, containing 23 and 22 exons, both encoding three transcript variants each, which can be translated into 872–890 aa-long proteins located on sterlet chromosome ARUT3 and on an unassigned scaffold, respectively.

The sterlet genome also encodes two paralogs of *ago4*: ago4_1 and ago4_2, both located on unassigned scaffolds. ago4_1 contains 10 exons and encodes a 437 aa-long protein. ago4_2 contains 18 exons and encodes three transcript variants, which can be translated into 839–875 aa-long proteins. As the spotted gar genome contains a single *ago4* encoding 873 aa protein, ago4_2 appears to be more conserved, and *ago4_1* represents a truncated gene, whose product contains the N-terminal domain of the sterlet ago4 proteins. It can be linked to the fact that *ago4_1* is a terminal gene, located at the end of the short unassigned scaffold NW_023263670.1 consisting of 23,000 bp and encoding only two genes (ago4_1 and claspin-like pseudogene). Thus, it may result from poor assembly and future improved assemblies will restore the full gene structure.

Both *ago1* and *ago3* genes have lost one of their paralogs in the sterlet genome and are represented by single copy genes. *ago1* consists of nine exons and encodes a 448 aa-long protein, while *ago3* has 19 exons and encodes two transcript variants, which can be translated into 867 and 860 aa proteins (XP_034771548.1, XP_034771549.1). Both *ago1* and *ago3* are located on the sterlet chromosome ARUT59.

Although sterlet has lost one of the *ago1* paralogs, both copies have been retained in the African clawed frog, Atlantic salmon, Rainbow trout, and goldfish. Surprisingly, four copies were found in the common carp, suggesting additional segmental duplications (Figure 9). Again, the presence of multiple paralogs affected the congruence of the phylogenetic tree.

A similar situation was observed for *ago3*, which is a single-copy gene in sterlet, but both paralogs have been retained in the African clawed frog, Atlantic salmon, and Rainbow trout. We found three paralogs of *ago3* in the common carp and five paralogs in the goldfish (Figure 10). *Danio rerio*, *Takifugu rubipes*, and *Oryzias latipes* retained two copies of the gene, probably resulting from teleost specific WGD.

*Ago4* has been retained in two copies in the sterlet, Atlantic salmon, Rainbow trout, and goldfish, but four copies were found in the common carp (Figure 11).

Expression analysis of *ago* genes in different organs of *Acipenser ruthenus* revealed that *ago2_1* is transcribed in testis, spleen, and muscle and at a lower level in undifferentiated gonads. *ago2_2* is not active in any of the investigated tissues (Figure 3). *ago4_1* is transcribed in almost all tissues except liver. *ago4_2* was found to be actively transcribed generally in muscle, ovary, testis, and undifferentiated gonads. Transcription of ago1 was observed primarily in brain, ovary, and undifferentiated gonads, while *ago3* transcripts are present in ovary, muscle, testis, and undifferentiated gonads (Figure 3).

### 2.7. Paralogs of Piwi-Like Proteins

The duplication of the *piwi-like1* argonaute gene also resulted from the acipenserid-specific WGD event. *piwil1_1* and *piwil1_2*, containing 22 and 21 exons and encoding 860 aa-long proteins, are located on the paralogous sterlet chromosomes ARUT12 and ARUT19, respectively. Both *piwi-like 1* copies are active and, as expected, transcribed primarily in undifferentiated gonads, ovary, and testis (Figure 3). This observation further validates the transcriptomic data used in this research, because piRNA and corresponding piwi proteins are mostly active in germline cells [19].

Phylogenetic analysis revealed the presence of two paralogs in tetraploid carps, *Xenopus laevis*, and, surprisingly, in the diploid *Anolis carolinensis*. In diploid teleosts and in tetraploid *Salmo salar* and *Oncorhynchus mykiss* only a single copy of the gene was found (Figure 12).

*piwil2_1* and *piwil2_2*, both containing 23 exons and encoding 1066 aa-long proteins, are located on the sterlet chromosomes ARUT41 and unplaced scaffold NW_023261183.1, respectively. These paralogs are expressed in gonads, as expected (Figure 2). Phylogenetic analysis indicated the presence of three paralogs in the tetraploid *Carassius auratus*, while other polyploid vertebrates retained only a single *piwil2* gene (Figure 13).

## 3. Discussion

### 3.1. Paralog Retention in Recent Polyploids

The sterlet reference genome analysis has led to an estimate for the duplicate retention rates of ≈70% after the acipenseridae-specific WGD (180 MYA), which is much higher than the about 15–20% ohnolog retention rate after the teleost WGD (320 MYA) [49]. It was shown that the sterlet genome is generally characterized by very slow evolutionary rates and it may serve as a useful representative of the conserved ancestral actinopterygian genome [11]. Our results further confirm this data, as we demonstrate that the majority of here investigated *Acipenser ruthenus* genes have retained both paralogs (Table 2) and both these copies are active. We have shown that most of the studied retained duplicates have a high degree of protein sequence homology (97–99%), with few exceptions (e.g., only 92% for *xpo5* paralogs) (Table 1). We found that only two studied miRNA-associated genes are represented in the sterlet genome by a single copy—*ago1* and *ago3*, while other polyploid vertebrates maintained at least two copies of *ago1* and *ago3*.

### 3.2. Ago Paralogs and Rediploidization by Segment Excision

Mechanisms of duplicate gene loss following WGD can be divided into structural and functional ones. Structural mechanisms are based on the duplicated genes deletion through random excision by the elimination of chromosomal segments containing one or more genes. Functional mechanisms are connected to epigenetic silencing and pseudogenization [50]. The fact that we did not find any degenerated pseudogenes for *ago1* and *ago3* may reflect the tendency of segment excision/deletion prevailing over step-by-step pseudogenization. Both, *ago1* and *ago3* are located on chromosome ARUT 59 (one of the small sterlet chromosomes that has lost its whole paralogous counterpart), further indicating a phenomenon of segmental rediploidization.

Previously, it was shown that in some vertebrates (from chicken to mammals) *Ago1*, *Ago3*, and *Ago4* genes are located on the same chromosome, forming the AGO gene cluster (evolutionarily conserved linkage group of the *Ago4-Ago1-Ago3* and several neighboring genes), while *Ago2* gene is located on a different chromosome [51,52]. The organization is conserved in the sterlet genome, where *ago1* and *ago3*, are located on chromosome ARUT59 and *ago2_1* on chromosome ARUT3. At the same time, *ago2_2*, *ago4_1*, and *ago4_2* paralogs have originated through acipenserid-specific WGD and were detected on unplaced scaffolds, which may result either from assembly errors or from post-WGD chromosomal rearrangements.

### 3.3. Dicer Paralogs: Discovery of an Ancient Segmental Duplication

Segmental duplications (SDs) are another crucial type of duplications in general and the only one for species that have not gone through the WGDs. Segmental duplications can be adjacent (tandem duplications), separated along a particular chromosome (intrachromosomal) or on distinct chromosomes (interchromosomal) [53]. An additional segmental duplication was demonstrated for the *dicer1*, resulting in four *dicer* copies in the sterlet genome. *dicerA_1* and *dicerB_1* result from an ancient 50 kb tandem segmental duplication (containing *clmn* and *dicer* genes), which preceded the acipenserid-specific WGD. *Dicer* family expansions via segmental duplications achieving four and more *Dicer* duplicates were previously demonstrated for placozoa and plants [31]. According to our data, the here reported *dicer1* containing segmental duplication, which we have identified, is the first case of four detected *Dicer* genes among vertebrates so far. A high level of gene structure conservation and detection of transcripts for all four *dicer* copies suggest that all of them may have retained a function. Although generally *dicerB* paralogs are transcribed more actively than *dicerA* paralogs, high transcription levels indicate that the coduplicated regulatory elements are also functionally conserved (Figure 3).

### 3.4. Xpo and Ago Genes: Variable Copy Number across Recent Polyploids

We noticed that both *Cyprinus carpio* and *Carassius auratus* have lost one paralog of *xpo5*, but that there was an expansion of *xpo1* of up to three and five paralogs, respectively. As xpo1 is responsible for many functions, including nuclear transport of proteins, snRNAs, mRNA, and in some species is involved in pre-miRNA processing and transport [54], we expect that different paralogs may be subjected to subfunctionalization.

*Cyprinus carpio* and *Salmo salar* have retained only a single paralog of *drosha* and *ago2*, while other polyploid vertebrates (*Acipenser ruthenus*, *Xenopus laevis*, and *Carassius auratus*) maintained both duplicated copies. These two genes encode proteins possessing catalytic activity and are critical for miRNA processing. It seems that the high degree of paralog similarity indicates the functionality of both retained copies. *ago2_1* was higher expressed than *ago2_2* indicating a possible suppression effect, leading to loss of function.

It is important to note that salmonids *Salmo salar* and *Oncorhynchus mykiss* have diverged only about 26-30 MYA [12], but *Oncorhynchus mykiss* maintained two *ago2* gene paralogs and surprisingly magnified the number of *xpo1* genes up to four copies, while *Salmo salar* lost one *ago2* copy and retains two *xpo1* gene copies.

*Acipenser ruthenus* retained a single copy of *ago1* and *ago3*, whereas other polyploid vertebrates retained both paralogs. Other loss-of-function cases caused by deletion/degeneration of one of paralogs were restricted to *Xenopus laevis Ago4* and *Salmo salar piwil1*.

Interestingly, the genomes of the diploid species *Xenopus tropicalis* and tetraploid *Xenopus laevis* contain three tandemly arranged *Xpo5* gene copies, resulting from segmental duplications, while the polyploid carps retained only a single copy. This may indicate that this gene is not dosage sensitive.

Although the ancestor of teleosts experienced a specific third round of whole genome duplication 320 MYA, *Danio rerio*, as well as other teleosts, is considered to be diploid, but its genome retained many paralogs resulting from the last whole genome duplication event. We noticed that zebrafish retained several duplicated genes such as *xpo1* and *ago3*. Moreover, our phylogenetic tree indicated that both paralogs of *xpo1* were present in the ancestor of carp before the carp-specific genome duplication event, as each of the *Danio rerio* paralogs forms a clade with the respective carp paralogs. However, *Salmo salar* appears to have lost one of these ancient teleost paralogs. These data indicate that microRNA processing genes seem to be prone to retention after whole genome duplication and segment duplication events and that teleosts are tolerant to additional *ago* and *xpo* gene copy accumulation.

### 3.5. Subgenome Dominance and Paralog Retention

Making the assumption that the phylogenetic gene trees generally reflect evolutionary distances well, we can make the conclusion that all protein products of *Acipenser ruthenus* duplicated genes studied diverged from each other approximately equally. This is not a common situation for all investigated polyploid vertebrates. Thus, the evolution of duplicates genes strongly depends on the type of polyploidization event: allopolyploidization (like in *Xenopus laevis*) might lead to subgenome dominance and massive gene loss and generation on one of the paralogs, while autopolyploidization (like in sterlet or salmonids) usually is accompanied by a random redundancy reduction on both paralogs.

It was shown previously that *Xenopus laevis* S and L subgenomes diverged from each other around 34 MYA and from *Xenopus tropicalis* around 48 MYA [10]. We observed that *Xenopus laevis* duplicated copies of *Dicer*, *Drosha*, *Xpo1*, and *Xpo5* diverged from each other and from orthologs from *Xenopus tropicalis* with different rates. We paid special attention to the fact that the L subgenome is more similar to ancestral variants and *Xenopus tropicalis*, while the S subgenome is much more dynamic [10]. It was demonstrated that the S subgenome also went through extensive intrachromosomal rearrangements resulting in large inversions (in chromosomes 2S–5S, and 8S), deletions, and shorter rearrangements [10]. We found that both *Xpo5* copies were derived from chromosome 5 of the S subgenome. Moreover, we revealed the presence of a third tandemly arranged truncated *Xpo5* copy in this subgenome. A similar situation (the presence of three tandemly arranged *Xpo5* copies) was observed in the diploid *Xenopus tropicalis* genome. This observation implies that the segmental duplication might have occurred before the divergence of *Xenopus laevis* and *Xenopus tropicalis* on the ancestral chromosome 5 and that the duplicated *Xpo5* copy from *Xenopus laevis* L subgenome was lost despite the conserved status of this subgenome. We found that both *DGCR8* copies (LOC108716057 and dgcr8.L) in *Xenopus laevis* were derived from a segmental duplication on the 1L chromosome. These copies are tandemly arranged and contain 16 and 11 exons, respectively. *Dicer/Drosha/Xpo1* copies were preserved in both subgenomes but the respective copies from the S genome (*Dicer1/Drosha/Xpo1*) appear to be highly evolved.

Although the salmonid WGD occurred more recently (about 80–100 MYA) than the Ac3R WGD, the products of duplicated genes (*dgcr8*, *dicer*, *xpo5*) are more diverged (only 89.47% (dicer), 90.04% (xpo5) 91.1% (dgcr8) of protein similarity compared to 92.40% (xpo5) and ≈99% (dgcr8 and dicer) of paralog similarity in sterlet). As in salmonids no subgenome dominance was observed [7], we suggest that the rediploidization process is much faster for several genes including *dicer* and *dgcr8* in salmonids than in sturgeons which agrees with the overall very low evolutionary rate reported for sterlet [11].

### 3.6. Hexanucleotide Expansion in Drosha Genes

We analyzed (GAGAGG)_n_ poly-hexanucleotide insertion in *drosha* genes not only in the reference sterlet genome assembly, but also in three other publicly available sterlet genome assemblies: fAciRut3.1 paternal and maternal haplotypes obtained at the Wellcome Sanger Institute (GCA_902713435.1 and GCA_902713425.1) and an assembly from Yangtze River Fisheries Research Institute (GCA_004119895.1). All studied sterlet genomes contain both *drosha* duplicated gene copies. We found that in all these assemblies one paralog (which we consider as *drosha1*) contains two regions with tandemly arranged hexanucleotides of similar sequence—GAGAAG and GAGAGG (except *drosha1* from GCA_902713435.1, which contains (GAGAGG)_3_) and the second derived paralog, *drosha2*, which contains a different number of GAGAGG-repeats: six in GCA_902713425.1, eight in GCA_004119895.1, nine in GCA_902713435.1, and 11 in GCF_010645085.1. (Figure 9), but no GAGAAG sequence.

Probably, the ancestral *drosha2* experienced a G/A substitution transforming GAGAAG to GAGAGG and thus forming a (GAGAGG)_2_ substrate for repeat expansion leading to different numbers of repeats in each sequenced sterlet genome.

It should be noted that *drosha2* is not only transcribed according to our transcriptomic data but also demonstrated higher activity in ovary, spleen, and undifferentiated gonads despite the presence of 11 hexanucleotide repeats in its coding sequence.

We analyzed this region in transcriptomes of several other representatives of Acipenseriformes to estimate the evolutionary dynamics of this hypervariable region in *drosha* genes. It revealed that transcripts from *Acipenser baerii*, *Acipenser sinensis*, *Acipenser gueldenstaedtii,* and *Acipenser sturio* hybrid sturgeon contain three tandemly arranged hexanucleotides—GAGAAG and (GAGAGG)_2_. In *Acipenser oxyrinchus* we found (GAGAAG)_2_ followed by GACAGG. In *Acipenser oxyrinchus* we also found GAGCGG and (GAGAGG)_2_, GAGAGG-GAGCGG-GAGAGG variants in this region.

We observed this hexanucleotide repeat expansion in the *Acipenser ruthenus drosha* cds regions corresponding to the arginine/serine-rich (RS-rich) domain in the human *Drosha* which shapes the N-terminal protein region together with the proline-rich (P-rich) domain, while Drosha catalytic activity is associated with its C-terminal end (two RNAse III domains and double-stranded RNA-binding domain) [55]. The central domain and C-terminal domains were pretty conservative, while N-terminal domains are more variable in human Drosha [56]. It was reported previously that the RS-rich domain is linked to cellular localization and protein stabilization because of its post-translational modifications such as phosphorylation and acetylation [57,58,59,60]. The loss of a part of the RS-rich domain during alternative splicing affected the specific subcellular localization (nuclear or cytoplasmic) of different *Drosha* transcripts and proteins without the loss of its catalytic activity [61]. Over the past decade, the Drosha functional repertoire expanded significantly from pre-miRNA processing to a wide range of functions such as transcriptional activation and termination, post-transcriptional control of RNA stability, alternative splicing, protection against genotoxic stresses, expression of retrotransposons and viruses, cell differentiation and its aberrant expression is associated with multiple cancer types [62]. As in acipenserids the hexanucleotide expansion occurred at the N-terminal domain, the high variability in copy number in one of the paralogs may reflect the neo- or subfunctionalization process, as it seems conserved across species for long evolutionary times (over 200 myr).

### 3.7. Ago2 and Its Potential Slicing Activity in the Sterlet

It is interesting to note that Ago2 is the only Argonaute protein of miRNA-class AGO that retains its RNA target cleavage activity in vertebrates. The sterlet genome encodes two full-length *ago2* genes obviously maintaining the slicing activity. At the same time, argonaute2-catalyzed miRNA slicing in most fish is impaired because of some mutations that emerged in the ancestor of most teleost fish [63]. As Acipenseriformes lineage is an outgroup to teleosts, two sterlet ago2 share the vertebrate ancestral consensus amino acids motif at the Ago2 PIWI domain, which underwent loss-of-function mutations in teleosts (Figure 14). It suggests the sterlet as a better potential model species for miRNA-associated research than teleost fish, which are hardly suitable for RNAi experiments in comparison with spotted gar, and other basal ray-finned fish groups.

### 3.8. Expression Analysis

Some of the studied genes were previously included in the list of human housekeeping genes (*Drosha* and *Xpo1*) and thus are expected to be expressed at a constant level in all cells, while *Dicer*, *DGCR8*, *Xpo5*, and *Agos* demonstrate tissue specificity [64].

Analysis of miRNA-associated genes expression in *Acipenser ruthenus* demonstrates that for some genes (*ago2*, *xpo5*, and *xpo1*) one paralogous copy is more transcriptionally active than the other. Ago2 is the only Argonaute protein that maintains its catalytic activity essential for miRNA maturation and Xpo5 and Xpo1 are involved in miRNA transport from the nucleus to the cytoplasm. These three proteins are essential for the metabolism of the large majority of miRNAs and their genes are expected to be expressed at high levels. Transcription analysis demonstrated that *ago2_1* is the most highly transcribed gene in the spleen (in comparison to other studied tissues), *xpo5_1* expression is quite high in all tested organs with local maximums in the spleen, testis, and the ovary, and *xpo1_2* is actively transcribed in the brain, the ovary, and testis. It is important to note that exportin-encoding duplicated copies (*xpo1_1* and *xpo5_2*) are also transcriptionally active, but they are transcribed at a lower level. *ago2_2* transcription is completely suppressed, indicating that only *ago2_1* is transcribed in *Acipenser ruthenus* maintaining its catalytic activity. This indicated functional deduplication.

*drosha* and *dgcr8* duplicated copies are transcribed approximately at the same level. *dicer* duplicates are transcriptionally active nearly at the same level with some fluctuations between gene copies and a maximum in the ovary for *dicerB*. *ago1* is the most highly transcribed in the ovary. *ago3* is transcribed approximately at the same level in all tested organs. *ago4_2* is more active than *ago4_1* in all investigated tissues except liver. Both paralogs of *piwil1* and *piwil2* were found to be the most transcriptionally active in undifferentiated gonads, and increased transcription was found in the testis and ovary. This result may be expected, because piwi proteins repress transposons and retroviruses both in germline cells during gametogenesis and in mature gonads (testis and ovary). Previously, experiments with another sturgeon species, *Acipenser dabryanus*, identified and characterized two piwi proteins (piwil1 and piwil2) and their expression in different tissues and organs. Our results on *piwil1* and *piwil2* expression are in agreement with the data on *Acipenser dabryanus*, where gonad specific transcription was demonstrated [65].

## 4. Materials and Methods

### 4.1. Retrieving Paralogs from Sterlet Databases

Protein sequences were identified using reference protein sequences of the sterlet from the GenBank Database (GCF_010645085.1) [11] used as a query in the BLASTP 2.10.1 algorithm [66] with default parameters against reference protein sequences corresponding to the open reading frames (ORFs) encoded in the investigated vertebrate genomes. A similar approach was applied for searching nucleotide sequences using the BLASTN algorithm with default settings.

### 4.2. Retrieving Orthologs from Databases

Orthologous proteins were derived from NCBI database using previously annotated sterlet proteins as a query in blastp against refseq_protein database with specified organism selected: (*Acipenser ruthenus* (GCF_010645085.1), *Carassius auratus* (GCF_003368295.1), *Cyprinus carpio* (GCF_000951615.1), *Xenopus laevis* (GCF_001663975.1), *Danio rerio* (GCF_000002035.6), *Salmo salar* (GCF_000233375.1), *Oncorhynchus mykiss* (GCF_013265735.2) *Takifugu rubripes* (GCF_901000725.2), *Oryzias latipes* (GCF_002234675.1), *Lepisosteus oculatus* (GCF_000242695.1), *Latimeria chalumnae* (GCF_000225785.1), *Xenopus tropicalis* (GCF_000004195.4), *Gallus gallus* (GCF_000002315.6), *Anolis carolinensis* (GCF_000090745.1), *Mus musculus* (GCF_000001635.27), and *Homo sapiens* (GCF_000001405.39)). These proteins were manually verified and only the longest transcript variant per gene was used for further analysis.

For verification of our results, we additionally used three other *Acipenser ruthenus* genome assemblies (GCA_902713435.1, GCA_902713425.1, GCA_004119895.1).

To investigate hexanucleotide expansion we used transcripts encoding Drosha protein from *Acipenser baerii* (GICB01032491.1, GICD01041616.1), *Acipenser sinensis* (GGYF01039448.1, GGYF01039449.1), *Acipenser gueldenstaedtii* (GGWK01368855.1), *Acipenser sturio* hybrid sturgeon (GGQL01028387.1, GGQL01028389.1, GGQL01028390.1, GGQL01028393.1, GGQL01028394.1, GGQL01028392.1), *Acipenser oxyrinchus* (GGZT01133858.1, GEUL01095986.1, GGZT01115826.1, GGZT01133857.1, GEUL01095987.1, GGWJ01002666.1, GGZX01632362.1). These transcripts were found by blastn of *drosha* mRNA against the Transcriptome Shotgun Assembly (TSA) database in GenBank.

### 4.3. Phylogenetic Analysis

Multiple sequence alignments were generated by MAFFT 7.471 and MUSCLE 3.8.31 tools with default settings [67,68]. The selection models for protein phylogenetic tree building were selected via Smart Model Selection (SMS) implemented in PhyML and presented in Table 3 for each tree [69]. Phylogenetic trees were rooted using reconciliation in Notung 2.9.1.5 software using the species tree obtained from the NCBI Taxonomy Browser (https://www.ncbi.nlm.nih.gov/Taxonomy/CommonTree/wwwcmt.cgi) [70,71,72]. Phylogenetic analysis was performed using PhyML 3.0 software with default settings and visualized using MEGA X, Dendroscope v3.5.10, and iTOL 5.6.3 [73,74,75,76]. A map of the chromosome regions containing a segmental duplication including *dicer* genes (Figure 5) was generated with genoPlotR [77].

### 4.4. Expression Analysis

Previously published RNA sequencing data of brain, liver, muscles, ovary, spleen, testis, and undifferentiated gonads [11] were used for gene expression analysis. Filtering by quality and adapter trimming was performed using fastp 0.20.0 [78] with the parameters “-3-5-detect_adapter_for_pe-c-g-l 50”. Trimmed reads were aligned using hisat2 2.2.0 [79] to the sterlet genome with standard settings. samtools 1.9 [80] with the “-q 30” option was used to filter alignments by quality (MAPQ > 30). For each tissue, potential transcripts were assembled using stringtie 2.1.4 [81] with standard settings. FPKM (Fragments Per Kilobase of transcript per Million mapped reads) for each transcript was calculated and written into a GTF file. To extract FPKM values for genes of interest, their coordinates were extracted from the genomic annotation and intersected with the GTF file using bedtools 2.27.1 [82] with the “intersect” option. To plot a heatmap, the FPKM values were transformed using the log_10_(1 + FPKM) formula. The plot was created in MATLAB 9.8.0.

## 5. Conclusions

Here we demonstrate that most microRNA processing genes retain two copies after the WGD events in most vertebrates. Some genes (*agos* and *exportins*) seem to be more tolerant to dosage and can additionally be amplified through segmental duplications. Slightly different expression patterns may indicate the nascent subfunctionalization of paralogous copies. Special attention should be paid to microsatellite repeat instability as a possible way of paralogous copy neo- or subfunctionalization. Generally, it seems that amplification of microRNA processing genes is a common evolutionary process and further neo- or subfunctionalization might be beneficial for different pathways of microRNA processing.

## Figures and Tables

**Figure 1 ijms-21-09562-f001:**
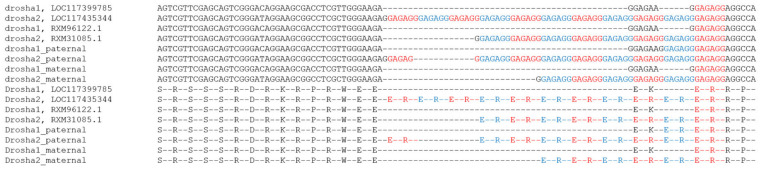
Alignment of *drosha* genes and corresponding proteins obtained from different sterlet genome assemblies detailing GAGAGG hexanucleotide amplification in the exon 4 of *drosha* genes. drosha1, LOC117399785 and drosha2, LOC117435344 are from the reference sterlet genome assembly (GCF_010645085.1); drosha1/drosha2 paternal—from GCA_902713435.1; drosha1/drosha2 maternal were retrieved from GCA_902713425.1; RXM96122.1 and RXM31085.1—from GCA_004119895.1 [48]. Repeats are highlighted with red and blue.

**Figure 2 ijms-21-09562-f002:**
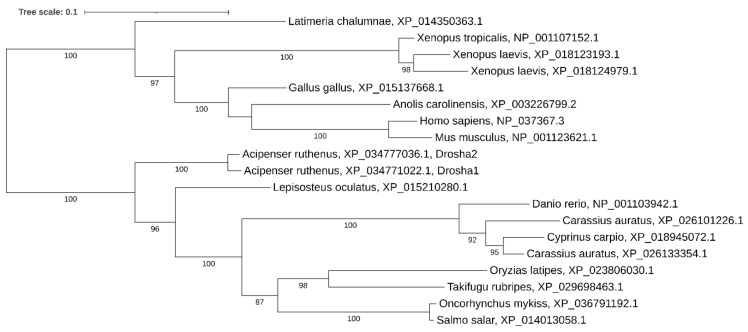
Phylogenetic tree of Drosha proteins in vertebrates.

**Figure 3 ijms-21-09562-f003:**
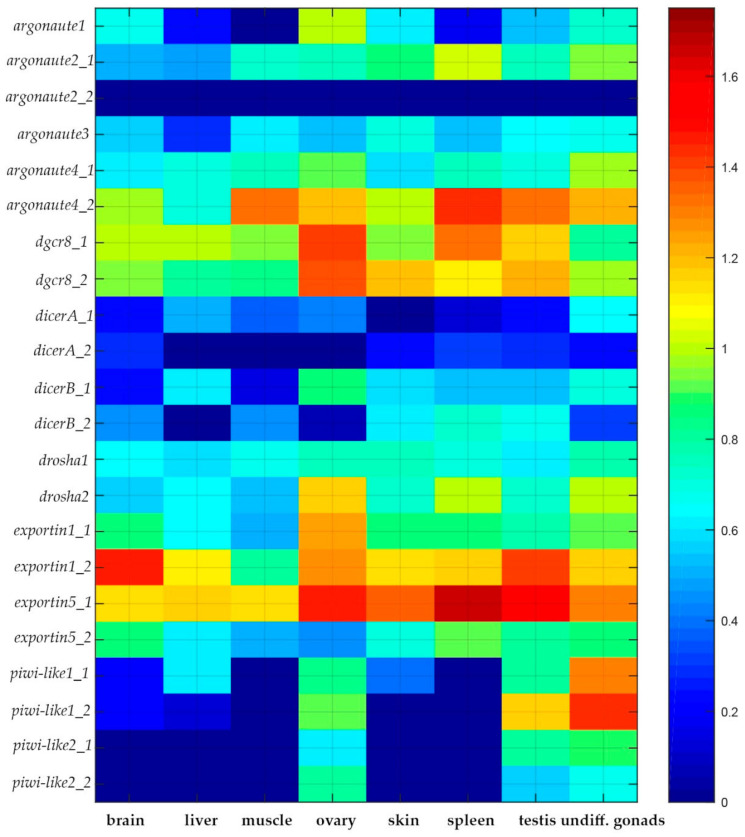
Heat map of expression values for microRNA biogenesis genes in different organs of *Acipenser ruthenus*. To plot a heatmap, the Fragments Per Kilobase of transcript per Million mapped reads (FPKM) values were transformed using the log_10_(1 + FPKM) formula.

**Figure 4 ijms-21-09562-f004:**
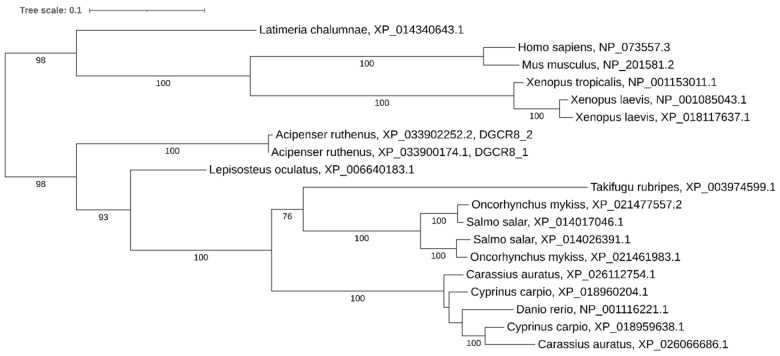
Phylogenetic tree of DGCR8 proteins in vertebrates.

**Figure 5 ijms-21-09562-f005:**
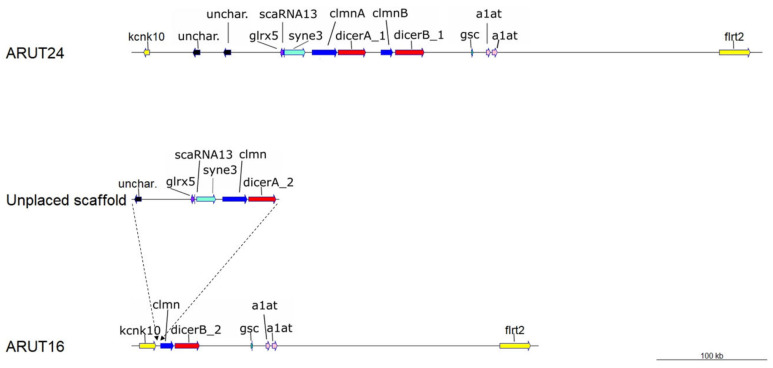
Map of the chromosome regions containing a segmental duplication including *clmn* and *dicer* genes. kcnk10—potassium channel, subfamily K, member 10; unchar.—uncharacterized protein; glrx5—glutaredoxin-related protein 5, mitochondrial-like; scaRNA13—small Cajal body-specific RNA 13; syne3—nesprin-3-like; clmn—calmin-like; gsc—homeobox protein goosecoid-like; a1at—alpha-1-antiproteinase-like; flrt2—leucine-rich repeat transmembrane protein flrt2-like. Scale bar: 100 kb. The hypothetical placement of the unplaced scaffold to the chromosome ARUT16 is clarified hereinafter in the text.

**Figure 6 ijms-21-09562-f006:**
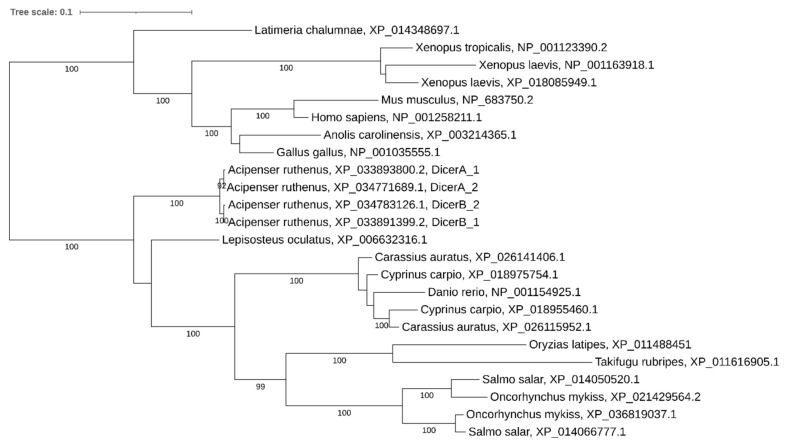
Phylogenetic tree of Dicer proteins in vertebrates.

**Figure 7 ijms-21-09562-f007:**
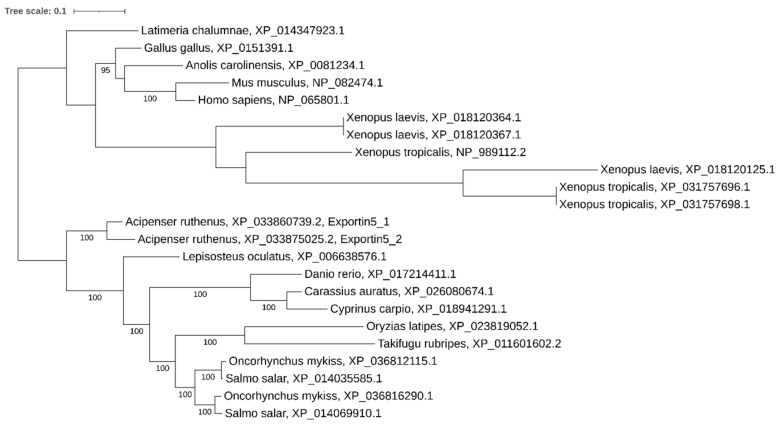
Phylogenetic tree of Exportin 5 proteins in vertebrates.

**Figure 8 ijms-21-09562-f008:**
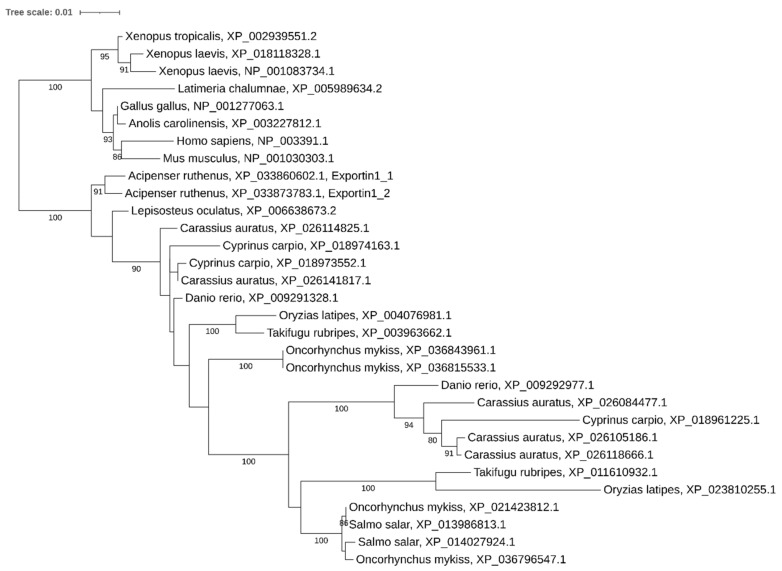
Phylogenetic tree of Exportin-1 proteins in vertebrates.

**Figure 9 ijms-21-09562-f009:**
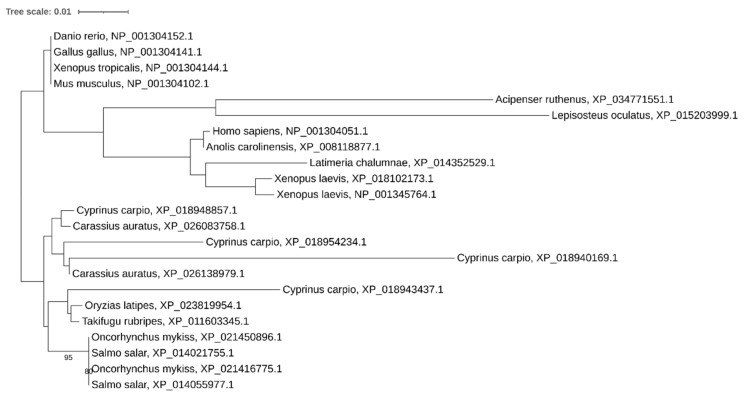
Phylogenetic tree of Argonaute1 proteins in vertebrates.

**Figure 10 ijms-21-09562-f010:**
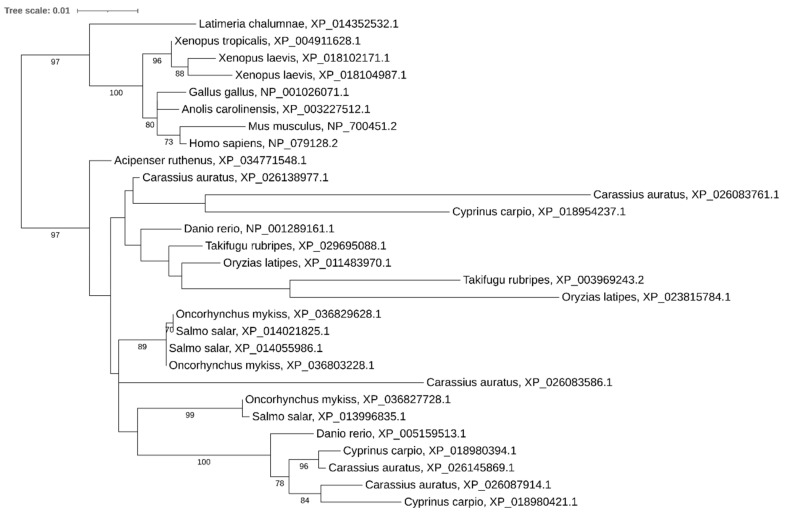
Phylogenetic tree of Argonaute3 proteins in vertebrates.

**Figure 11 ijms-21-09562-f011:**
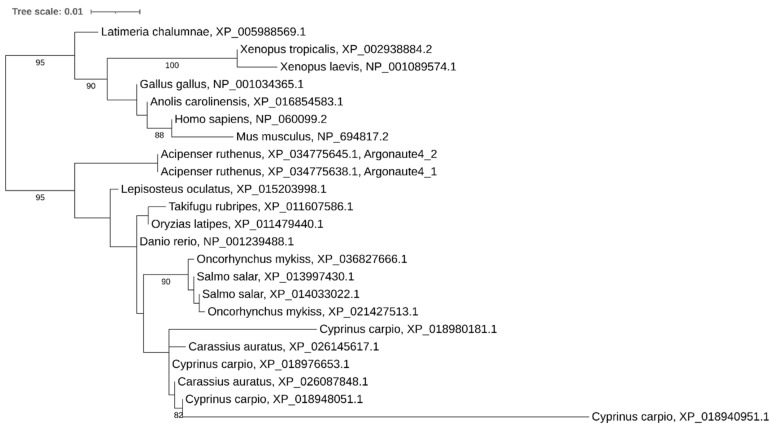
Phylogenetic tree of Argonaute4 proteins in vertebrates.

**Figure 12 ijms-21-09562-f012:**
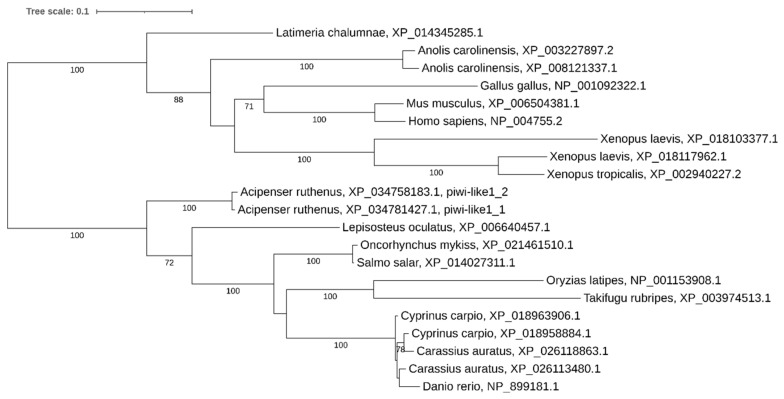
Phylogenetic tree of piwi-like1 proteins in vertebrates.

**Figure 13 ijms-21-09562-f013:**
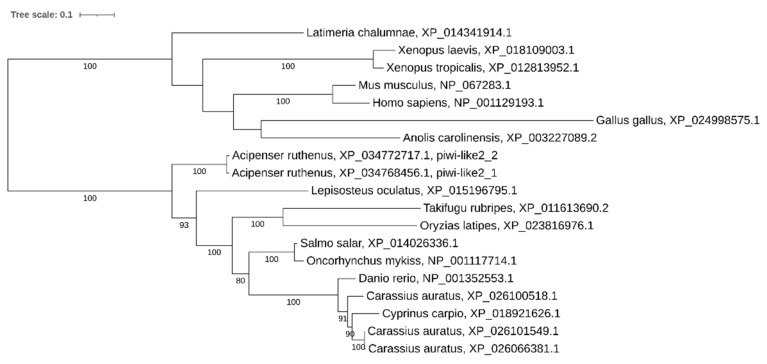
Phylogenetic tree of piwi-like2 proteins in vertebrates.

**Figure 14 ijms-21-09562-f014:**
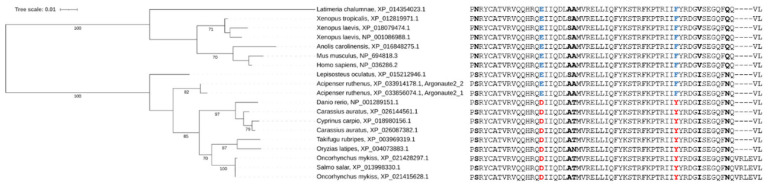
Comparative sequence analysis of Ago2 and its orthologs in vertebrate species including polyploids. The phylogenetic tree shows evolutionary distances between both sterlet ago2 protein paralogs and vertebrate orthologs (left). The sequence alignment highlights differences within a short region of the Ago2 PIWI domain (right). All residues that vary among studied species are in bold. The two substitutions that were supposed to be linked with the loss of ago2 catalytic activity (slicing) [63] are indicated by red letters, while blue letters indicate that these ago2 orthologs are active in corresponding species.

**Table 1 ijms-21-09562-t001:** *Acipenser ruthenus* genes involved in miRNA biogenesis.

Gene Name	Gene ID	Scaffold Number ^1^	Number of Exons	Gene Length, bp	CDS Length, bp	Accession Number in the NCBI Protein Database	Protein Length, aa	Protein Coverage ^2^, %	Protein Identity, %
*drosha1*	117399785	3	32	50,094	3984	XP_034771022.1	1327	100	96.88
*drosha2*	117435344	4	33	46,432	4035	XP_034777036.1	1344		
*dgcr8_1*	117426576	12	14	9918	2388	XP_033900174.1	795	100	99.62
*dgcr8_2*	117427952	19	14	10,349	2388	XP_033902252.2			
*dicerA_1*	117422667	24	29	25,425	5685	XP_033893800.2	1894	100	99.89
*dicerA_2*	117968189	Unplaced scaffold	29	25,366	5685	XP_034771689.1			
*dicerB_1*	117421295	24	28	26,733	5691	XP_033891399.2	1896	100	99.79
*dicerB_2*	117973855	16	28	22,595	5691	XP_034783126.1			
*exportin5_1*, *xpo5_1*	117403025	5	32	25,720	3630	XP_033860739.2	1209	100	92.40
*exportin5_2*, *xpo5_2*	117411517	6	35	30,397	3630	XP_033875025.2			
*exportin1_1*, *xpo1_1*	117402968	5	25	26,506	3216	XP_033860602.1	1071	100	99.16
*exportin1_2*, *xpo1_2*	117410930	6	26	35,355	3216	XP_033873783.1			
*argonaute1*, *ago1*	117413876	59	9	5366	1347	XP_034771551.1	448		
*argonaute2_1*, *ago2_1*	117400353	3	23	48,512	2673	XP_033856074.1	890	100	99.66
*argonaute2_2*, *ago2_2*	117435258	Unplaced scaffold	22	46,489	2673	XP_033914178.1			
*argonaute3*, *ago3*	117413875	59	19	16,232	2604	XP_034771548.1	867		
*argonaute4_1*, *ago4_1*	117971566	Unplaced scaffold	10	8446	1314	XP_034775638.1	437	98	99.54
*argonaute4_2*, *ago4_2*	117413873	Unplaced scaffold	18	14,817	2625	XP_034775645.1	874		
*piwi-like1_1*	117426896	12	22	45,126	2583	XP_034781427.1	860	100	99.19
*piwi-like1_2*	117428443	19	21	16,714	2583	XP_034758183.1			
*piwi-like2_1*	117397939	41	23	12,687	3201	XP_034768456.1	1066	100	98.87
*piwi-like2_2*	117968944	Unplaced scaffold	23	12,655	3201	XP_034772717.1			

Scaffold number ^1^ equals chromosome number [11]; protein coverage ^2^—the aligned length of the total length of the larger protein compared to the shorter paralogous protein; bp—base pair, aa—amino acid.

**Table 2 ijms-21-09562-t002:** The number of miRNA-associated genes in different vertebrate species.

Species	Ploidy	Number of Gene Copies and the Type of Duplication Origin										
		*drosha*	*dgcr8*	*dicer*	*xpo1*	*xpo5*	*ago1*	*ago2*	*ago3*	*ago4*	*piwil1*	*piwil2*
*Acipenser ruthenus*	4n	2 WGD	2 WGD	4 SD, WGD	2 WGD	2 WGD	1	2 WGD	1	2 WGD	2 WGD	2 WGD
*Anolis carolinensis*	2n	1	-	1	1	1	1	1	1	1	2	1
*Carassius auratus*	4n	2 WGD	2 WGD	2 WGD	5 SD, WGD	1	2 WGD	2 WGD	5 WGD, SD	2 WGD	2 WGD	3 WGD
*Cyprinus carpio*	4n	1	2 SD	2 WGD	3 SD, WGD	1	4 SD, WGD	1	3 SD, WGD	4 WGD	2 WGD	1
*Danio rerio*	2n	1	1	1	2 WGD	1	1	1	2 WGD	1	1	1
*Gallus gallus*	2n	1	-	1	1	1	1	-	1	1	1	1
*Homo sapiens*	2n	1	1	1	1	1	1	1	1	1	1	1
*Latimeria chalumnae*	2n	1	1	1	1	1	1	1	1	1	1	1
*Lepisosteus oculatus*	2n	1	1	1	1	1	1	1	-	1	1	1
*Mus musculus*	2n	1	1	1	1	1	1	1	1	1	1	1
*Oncorhynchus mykiss*	4n	1	2 WGD	2 WGD	4 WGD	2 WGD	2 WGD	2 WGD	3 WGD	2 WGD	1	1
*Oryzias latipes*	2n	1	-	1	2 WGD	1	1	1	2 WGD	1	1	1
*Salmo salar*	4n	1	2 WGD	2 WGD	2 WGD	2 WGD	2 WGD	1	3 WGD	2 WGD	1	1
*Takifugu rubripes*	2n	1	1	1	2 WGD	1	1	1	2 WGD	1	1	1
*Xenopus laevis*	4n	2 WGD	2 SD	2 WGD	2 WGD	3 SD	2 WGD	2 WGD	2 WGD	1	2 WGD	1
*Xenopus tropicalis*	2n	1	1	1	1	3 SD	1	1	1	1	1	1

WGD—duplicated copies were derived from whole genome duplication; SD—duplicated copies were derived from segmental duplication.

**Table 3 ijms-21-09562-t003:** Identified selection models for each constructed protein phylogenetic tree.

Protein	Model	Protein	Model
Drosha	JTT +G + I + F	Ago2	JTT + G + I
Dgcr8	JTT +G	Ago3	JTT + G + I
Dicer	JTT +G + I + F	Ago4	JTT + G + I
Xpo5	JTT +G + F	Piwi-like1	LG + G + I + F
Xpo1	JTT + G + I + F	Piwi-like2	JTT + G + I + F
Ago1	JTT + G + I + F		

JTT—Jones–Taylor–Thornton model; LG—Le-Gascuel model; +G—substitution rate heterogeneity across sites according to a gamma distribution; +I—the proportion of invariable sites; +F—indicates that amino acid frequencies can be modeled.

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
