# Peer review of "Evolution of MicroRNA Biogenesis Genes in the Sterlet (Acipenser ruthenus) and Other Polyploid Vertebrates"

_ijms, 2020, doi:10.3390/ijms21249562_

Round 1

Reviewer 1 Report

The proposed study contributes to better understanding of microRNA-associated gene evolution. Study of gene duplication in Acipenseridae is a challenge because of several whole genome duplication events occurred and Acipenserid karyotypes bear one of highest chromosome numbers within vertebrates. I have not found any fundamental issues in proposed manuscript. The ms is technically sound, presented in an intelligible fashion and written in good-quality English.

I have only few minor suggestions such as:

1) Introduction part seems to be too long which is prone to be less striking and interesting. I would prefer to shorten this part.

2) Lines 352-3: the sentence "Our results are in good agreement with the previously obtained data." There is not clear what "previously obtained data" means. Try to specify or delete this sentence.

3) Abbreviations and whole Scientific names are not correctly used within the ms (e.g., Danio rerio vs. D. rerio etc.).

Author Response

Dear Reviewer 1,

Thank you for reviewing our manuscript and valuable comments on our paper. We have carefully considered all suggestions and remarks and made the respective corrections throughout the manuscript.

Point 1: Introduction part seems to be too long which is prone to be less striking and interesting. I would prefer to shorten this part.

Response 1: We tried to shorten this part, but we kept the information which is criucial for understanding and interpretation of our results.

Point 2: Lines 352-3: the sentence "Our results are in good agreement with the previously obtained data." There is not clear what "previously obtained data" means. Try to specify or delete this sentence.

Response 2:   We re-phrased the sentence.

Point 3: Abbreviations and whole Scientific names are not correctly used within the ms (e.g., Danio rerio vs. D. rerio etc.).

Response 3: All used within the manuscript scientific names were corrected to the unified format containing only its full latin species names (e.g., Danio rerio).

Best regards,

Mikhail Fofanov

Reviewer 2 Report

"Evolution of microRNA biogenesis genes in the sterlet (Acipenser ruthenus) and other polyploid vertebrates" by Fofanov et al. presents a characterization of microRNA processing genes in this sturgeon lineage after a WGD event. The study is very descriptive, but is what it is and will probably generate some interest and citations in the literature. I have a few minor suggestions to improve the presentation and more importantly, the robustness of the methods.

In Table 1, there are a couple of corrections needed.

-"exones"->"exons"

-"Protein homology %"->"Protein identity %"; They are all homologous (descended from a common ancestor)

All trees throughout the paper should be rooted using reconciliation (e.g. Notung is a user friendly program that will enable this, but there are many implementations of this). From this a compilation of results can be presented as an image over the species tree.

Fig. 3 should be regenerated in a phylogenetically robust manner. The method of Rohlfs and Nielsen can be employed to do this.

For the previous two comments, Biol. Proc. Online 18:11 may be helpful in organizing methodological improvements, which describes a similar set of analyses for Atlantic salmon.

In the discussion, differences in the symmetry of evolution (and subgenome dominance) are thought to be related to if the origin of the WGD is an autopolyploidization event (like 3R, salmon) or an allopolylploidization event (like in Xenopus). This should be added to the discussion.

In the methods section for PhyML tree building, the model selection procedure and the identified models for each tree should be described.

Overall, this is a well written, but very descriptive, paper that will generate some interest in the literature in characterizing lineage-specific gene content.

Author Response

Dear Reviewer 2,

Thank you for reviewing our manuscript and valuable comments on our paper, which we used to improve the manuscript a lot. We have carefully considered all suggestions and remarks and made the respective corrections throughout the manuscript.

Point 1: In Table 1, there are a couple of corrections needed.

-"exones"->"exons"

-"Protein homology %"->"Protein identity %"; They are all homologous (descended from a common ancestor)

Response 1: Indicated corrections were made in the Table 1 (lines 165-167).

Point 2: All trees throughout the paper should be rooted using reconciliation (e.g. Notung is a user friendly program that will enable this, but there are many implementations of this). From this a compilation of results can be presented as an image over the species tree.

Response 2: All trees throughout the paper were rooted using reconciliation in Notung software using the species tree obtained from the NCBI Taxonomy Browser. This information was added to Materials and Methods section (line 578-580)

Point 3: Fig. 3 should be regenerated in a phylogenetically robust manner. The method of Rohlfs and Nielsen can be employed to do this.

Response 3: Fig.3 demonstrated a heat map of expression values for microRNA biogenesis genes in different organs of only single species (Acipenser ruthenus), neither other polyploid or diploid vertebrates. We have also only one transcriptome experiment replica done; experiment expansion can take more time to regenerate this figure in a phylogenetic robust manner.

Point 4: For the previous two comments, Biol. Proc. Online 18:11 may be helpful in organizing methodological improvements, which describes a similar set of analyses for Atlantic salmon.

Response 4: We have read Biol. Proc. Online 18:11 article (Extracting functional trends from whole genome duplication events using comparative genomics. 2016. https://doi.org/10.1186/s12575-016-0041-2) to find out how to organize methodological improvements in our article. Thanks to your review and advice to try reconciliation in Notung, we are getting closer to using similar methods only implemented in different programs. We obtained gene families through blast against protein reference sequences and following leaving only one protein sequence per gene. We used the same multiple alignment approaches (MAFFT and MUSCLE) and phylogenetic tree building (PhyML). Unlike this article, we used Smart Model Selection (SMS) for model selection and Notung for rooting and reconciliation instead Prottest and Softparsmap, respectively.

Point 5: In the discussion, differences in the symmetry of evolution (and subgenome dominance) are thought to be related to if the origin of the WGD is an autopolyploidization event (like 3R, salmon) or an allopolylploidization event (like in Xenopus). This should be added to the discussion.

Response 5: We added this information as required.

Point 6: In the methods section for PhyML tree building, the model selection procedure and the identified models for each tree should be described.

Response 6: This information was added to 4.3. Phylogenetic analysis in Materials and Methods (lines 576-577) and Table 3 (lines 585-589).

Best regards,

Mikhail Fofanov

Round 2

Reviewer 2 Report

The authors have satisfactorily addressed my concerns and this study is appropriate for publication.